**Data Availability Statement:** All relevant data are within the manuscript and its Supporting Information files.

# Lack of APOL1 in proximal tubules of normal human kidneys and proteinuric *APOL1* transgenic mouse kidneys

Natalya A. Blessing[1], Zhenzhen Wu[2], Sethu M. Madhavan[3], Jonathan W. Choy[4], Michelle Chen[4], Myung K. Shin[4], Maarten Hoek 🄳[5], John R. Sedor[2,6,7], John F. O'Toole[2,6], Leslie A. Bruggeman 🄳[2,6]*

1 Rammelkamp Center for Education and Research, MetroHealth Medical Center, Case Western Reserve University School of Medicine, Cleveland, OH, United States of America, 2 Department of Inflammation & Immunity, Cleveland Clinic, Case Western Reserve University School of Medicine, Cleveland, OH, United States of America, 3 Department of Medicine, Ohio State University, Columbus, OH, United States of America, 4 Merck & Company, Inc., Kenilworth, NJ, United States of America, 5 Maze Therapeutics, South San Francisco, CA, United States of America, 6 Department of Nephrology, Cleveland Clinic, Case Western Reserve University School of Medicine, Cleveland, OH, United States of America, 7 Department of Physiology & Biophysics, Case Western Reserve University School of Medicine, Cleveland, OH, United States of America

* bruggel@ccf.org

## Abstract

The mechanism of pathogenesis associated with *APOL1* polymorphisms and risk for non-diabetic chronic kidney disease (CKD) is not fully understood. Prior studies have minimized a causal role for the circulating APOL1 protein, thus efforts to understand kidney pathogenesis have focused on *APOL1* expressed in renal cells. Of the kidney cells reported to express APOL1, the proximal tubule expression patterns are inconsistent in published reports, and whether APOL1 is synthesized by the proximal tubule or possibly APOL1 protein in the blood is filtered and reabsorbed by the proximal tubule remains unclear. Using both protein and mRNA *in situ* methods, the kidney expression pattern of *APOL1* was examined in normal human and *APOL1* bacterial artificial chromosome transgenic mice with and without proteinuria. *APOL1* protein and mRNA was detected in podocytes and endothelial cells, but not in tubular epithelia. In the setting of proteinuria, plasma APOL1 protein did not appear to be filtered or reabsorbed by the proximal tubule. A side-by-side examination of commercial antibodies used in prior studies suggest the original reports of APOL1 in proximal tubules likely reflects antibody non-specificity. As such, *APOL1* expression in podocytes and endothelia should remain the focus for mechanistic studies in the APOL1-mediated kidney diseases.

## Introduction

Polymorphisms in the *APOL1* gene contribute significant risk for several forms of non-diabetic chronic kidney disease (CKD) [1–3]. This risk arises from a combination of recessive

**Funding:** This study was supported by the National Institutes of Health (www.nih.gov) grants DK108329 (LAB, JRS, JFO), DK095832 (JRS, JFO, LAB), and training grant DK007470 (NAB, JRS). The NIH had no role in study design, data collection and analysis, decision to publish, or preparation of the manuscript. Merck and Company provided support in the form of salaries and laboratory resources for authors (JWC, MC, MKS, MH), but did not have any additional role in the study design, data collection and analysis, decision to publish, or preparation of the manuscript. Maze Therapeutics provided support in the form of salary for the author (MH) but did not have any additional role in the study design, data collection and analysis, decision to publish, or preparation of the manuscript. The specific roles of these authors are stated in the author contributions section.

**Competing interests:** We have read the journal's policy and the authors of this manuscript have the following competing interests: The authors MH, JC, MC, and MKS were employees of Merck & Company, Inc. during the conduct of this study. There are no patents, products in development or marketed products to declare. The author MH was an employee of Maze Therapeutics during the conduct of this study. There are no patents, products in development or marketed products to declare. This does not alter our adherence to PLOS ONE policies on sharing data and materials.

inheritance of variant *APOL1* alleles plus exposure to an environmental stressor. The pathogenic function of the *APOL1* variants and how they interact with the environmental stressor to cause CKD are not fully understood. Although APOL1 is constitutively present in the circulation, prior studies have minimized a causal role for the circulating APOL1 protein [4–7], and efforts to understand kidney pathogenesis have focused on APOL1 expressed in renal cells. The *APOL1* kidney expression pattern remains unclear with published discrepancies between immunohistochemistry and mRNA *in situ* hybridization results, most notably the abundant APOL1 protein observed in the proximal tubule epithelium [8–10]. Since APOL1 is abundant in blood, it is unclear if APOL1 is filtered, especially in the setting of proteinuria, which could result in APOL1 protein reabsorption by the proximal tubule. Appearance of APOL1 in the proximal tubule, either by gene expression or reabsorption from filtrate, would indicate a potentially important role of the proximal tubule in APOL1-associated CKD pathogenesis.

APOL1 in circulation is bound to a 500 kDa $HDL_3$ particle, known as trypanolytic factor 1, a 1000 kDa lipid-poor IgM complex, known as trypanolytic factor 2, and possibly other lipid-poor, high molecular complexes associated with complement factors [7, 11–13]. The proteins produced by the two CKD-associated *APOL1* variant alleles, G1 and G2, bind the high molecular weight trypanolytic factors similar to the common allele G0 [14]. Although the APOL1 protein (42.5 kDa) is small enough to pass the glomerular filtration barrier size restriction limit, it is not known to circulate independent of these high molecular weight complexes [15]. However, lipoproteins and other components of HDLs can be filtered [16], and in the setting of proteinuria, larger molecular weight proteins normally restricted by the filtration barrier may appear in filtrate. It is unclear whether APOL1 or APOL1-containing complexes may be filtered in the setting of proteinuria.

To resolve these issues, we examined both *APOL1* gene and protein expression in human kidney tissue and kidneys from humanized transgenic mouse models that recreate native human *APOL1* expression. For these studies we validated commercial anti-APOL1 antibodies for specificity which may have contributed to prior discrepancies on kidney expression patterns. In addition, *APOL1* transgenic mouse models were made proteinuric by intercrossing with a model of HIV-associated nephropathy (HIVAN), a CKD strongly associated with carriage of *APOL1* risk alleles, to determine if proteinuria would change the appearance of APOL1 protein in tubular epithelial.

## Materials and methods

### Human tissue and mouse models

Formalin-fixed, paraffin-embedded human kidney (n = 4) and liver (n = 3) tissue from normal margins of cancer resections were obtained from the Cleveland Clinic Lerner Research Institute Biorepository. Three transgenic mouse lines expressing a 47 kb human genomic fragment in a bacterial artificial chromosome (BAC) encompassing the promoter and coding regions of the human *APOL1* gene for each G0, G1, or G2 alleles have been previously described [17, 18]. Each of the BAC-APOL1 transgenic lines were ≥10 generations backcrossed to FVB/Nj, a genetic background susceptible to HIVAN. The mouse HIVAN model used to induce proteinuria was the Tg26 *HIVAN4* congenic [19] that develops proteinuria and progressive focal segmental glomerulosclerosis as the parental Tg26 model (Jackson Laboratory #22354) but disease progression is slower. For all studies, kidney disease was monitored weekly after weaning by measuring proteinuria (i.e. amount of protein in spontaneously voided urine) by urinalysis using diagnostic dipsticks (Uristix, Siemens Healthcare). The IACUC-approved humane endpoint for kidney disease in this model was proteinuria reaching 4+ on dipstick. No animal reached this humane endpoint before the

predetermined study endpoint, as this study required an early stage of renal dysfunction (proteinuria on dipstick of 2+ to 3+) as functioning kidneys still capable of filtration and reabsorption were needed. All animals maintained normal weights and exhibited typical grooming and activity levels, and were not in pain or distress during the study. All animals were monitored twice a week by veterinary technicians not associated with this study for overall health and well-being, and no animals required analgesics or other supportive care. Each BAC-*APOL1* transgenic mouse line was intercrossed with the *HIVAN4* mouse model and were sacrificed at 8–10 weeks of age when their kidney disease progressed to protein-uria levels of ≥2+ by urine dipstick (G0 n = 15, G1 n = 11, G2 n = 9). Terminal urine, blood, and tissue collections were performed under deep isoflurane anesthesia followed immedi-ately (while still under anesthesia) by euthanasia using cervical dislocation. Albuminuria was assessed by polyacrylamide gel electrophoresis of 1μl urine, followed by Coomassie staining. Use of human tissue was reviewed and approved by the Cleveland Clinic IRB (IRB-06-050). Informed consent was waived because tissue was considered discard and no identifiable data were collected. All animal studies were reviewed and approved by Institu-tional Animal Care and Use Committees at the Cleveland Clinic and Case Western Reserve University (IACUC-2430).

## Tissue immunofluorescence

Formalin-fixed, paraffin-embedded kidney tissue sections were subjected to antigen retrieval as previously described [8]. Antibodies used to examine APOL1 expression in this paper was a rabbit anti-human APOL1 (Sigma, HPA018885, lot E105260, 1:400 dilution). Numerous lots of this Sigma rabbit polyclonal in addition to other commercial monoclonal antibodies were also evaluated and the results are summarized in **Supplemental Table 1A-1C and Supplemental Figure 1 in S1 Data.** Of note, many of these polyclonal antibody lots have been exhausted and are no longer available from Sigma. Other primary antibodies include: rabbit anti-mouse APOA1 (ThermoFisher, 1:500 dilution), goat anti-mouse CD31 (R&D Systems, 1:200 dilution), guinea pig anti-nephrin (USB, 1:200), and mouse anti-GLEPP1 (gift of Roger Wiggins, 1:50). FITC-labeled *Lotus tetragonolobus* lectin (Vector labs) was used to label proximal tubule cells as previously described [8]. For testing of com-mercial antibodies against human APOL1, kidney tissues were fixed using a variety of meth-ods and paraffin embedded for immunohistochemistry using various antigen retrieval methods. Details for each of these processing methods are provided in the **Supplemental Detailed Methods in S1 Data**.

## *APOL1* gene and protein expression

Plasma APOL1 protein concentration was determined using the Meso Scale Discovery electro-chemiluminescence immunoassay as described previously [5]. Concentration was determined relative to a known liquid chromatography-mass spectrometry calibrated human high density lipoprotein solution. Serum APOL1 and APOA1 protein levels were compared using Western blotting as previously described [20].

## mRNA in situ hybridization

*APOL1* gene expression was examined in mouse and human kidney and liver tissue using mRNA *in situ* hybridization. The manual RNAScope *in situ* hybridization kit (ACDBio) was used for formalin-fixed, paraffin-embedded tissue following kit instructions for either single probe or dual probe detection. Probes included human *APOL1* (catalog number 439871), murine nephrin (*Nphs1*, catalog number 433571), and murine CD31 (*Pecam1*, catalog number

316721). Pretreatments were 15 minute boiling and 30 minute protease digestion. The *in situ* hybridization signal appears as dots; one dot per ten cells is expected background.

## Results

Several commercial anti-human APOL1 antibodies (**Supplemental Table 1A in S1 Data**) were examined for specificity to human APOL1 using tissue or protein extracts from human and mouse kidneys and cells. Since mice do not have an ortholog of human *APOL1*, murine cells and tissues should not be immunoreactive to antibodies against human APOL1. In Western blotting, most commercial monoclonal and polyclonal antibodies were able to detect APOL1, although several weakly detected APOL1 with stronger detection of non-specific proteins (gels bands that did not coincide with the molecular weight of APOL1, **Supplemental Table 1B in S1 Data**). Most antibodies detected additional non-specific proteins, which differed depending on the source of protein extracts (**Supplemental Table 1B in S1 Data**). Some of these non-specific gel bands could be eliminated with pretreatment of protein extracts with deglycosylating enzymes (not shown) suggesting epitope recognition was dependent on glycans. In immunohistochemistry testing, wild-type mouse kidney used as a negative control was compared with APOL1 transgenic mouse kidney (**Supplemental Table 1C in S1 Data**). Many of the anti-APOL1 antibodies erroneously detected proteins in wild-type mouse kidney, including very strong immunostaining in tubules (**Supplemental Figure 1 in S1 Data**). Based on these validation studies, we selected a Sigma polyclonal antibody lot with limited off-target detection to examine APOL1 expression patterns.

In human kidney, APOL1 protein was detected by immunofluorescence in glomeruli but not tubules (**Fig 1A**). Similar expression patterns in human kidney were observed using mRNA *in situ* hybridization. APOL1 mRNA was present in glomeruli, peritubular capillaries, and larger vessels of the kidney, but not in any tubule segment (**Fig 1B**). A prior study of human liver transplant recipients established that circulating APOL1 protein is largely produced by the liver [21]. In human liver, APOL1 protein and mRNA expression patterns were similar, with expression detected in hepatocytes and vascular endothelia. In mice, *APOL1* expression was qualitatively lower in zone 1 and 2 hepatocytes compared to zone 3 hepatocytes, whereas in human, hepatocytes in all three zones were similar in expression level (**Supplemental Figure 2 in S1 Data**). The mouse liver tissues were from healthy adults, whereas the human liver tissues were normal margins from cancer resections that also had histopathologic evidence of steatosis, potentially contributing to this difference.

The observed expression pattern in human tissue was confirmed in three transgenic mouse lines expressing a human BAC encompassing the entire *APOL1* genomic region for either the G0, G1, or G2 alleles [17, 18]. APOL1 protein was abundant in podocytes and also was present in endothelial cells of glomerular capillaries, peritubular capillaries, and endothelia of larger vessels (**Fig 2A**). No APOL1 protein was detected in the proximal tubule or any other tubular segment (**Fig 2B**). Also similar to humans, the BAC-APOL1 transgenic mice had abundant APOL1 in blood (**Fig 3**), and expressed APOL1 protein and mRNA in liver hepatocytes (**Supplemental Figure 2 in S1 Data**). In the kidney, this circulating APOL1 protein also could be detected in blood trapped in vascular spaces (**Fig 2B**). There was no difference in the *APOL1* expression patterns between the APOL1-G0, -G1, or -G2 expressing mice (**Fig 2B**), and is consistent with previous studies in human biopsies from patients with different *APOL1* genotypes [8, 10, 22]. The APOL1 protein expression patterns also were confirmed using mRNA *in situ* hybridization (**Fig 4**). All three *APOL1* genotypes were examined but there were no differences based on *APOL1* genotype. Consistent with human kidney (**Fig 1**), APOL1 was expressed in podocytes (cell type confirmed with co-labeling with *Nphs1*) and vascular endothelia (cell type

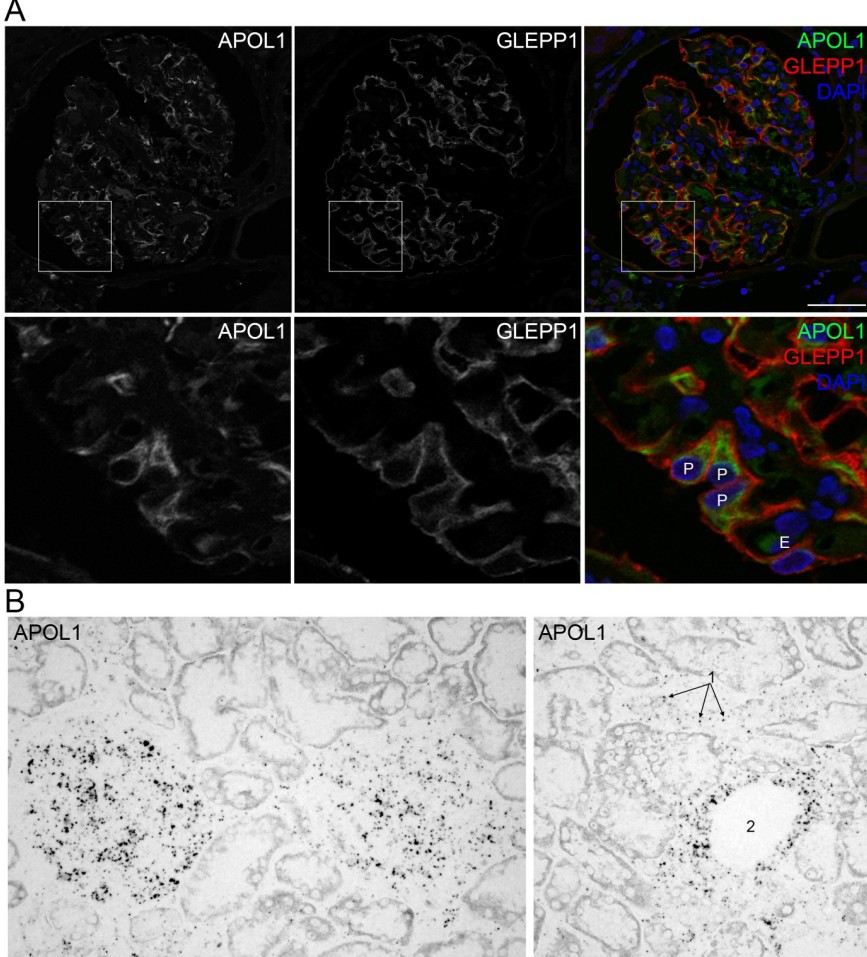

**Fig 1. APOL1 protein and mRNA expression in human kidney. A**. APOL1 protein expression by immunofluorescence microscopy using validated anti-APOL1 antibodies, with co-immunostaining with GLEPP1 to identify podocytes and DAPI as a nuclear stain. **B**. APOL1 mRNA expression by *in situ* hybridization. Expression was evident in podocytes ("P") and vascular endothelia of glomerular capillaries ("E"), peritubular capillaries ("1", arrows), and larger vessel ("2") epithelia, but not tubular epithelia. Scale bar = 40μm.

confirmed with co-labeling with *Pecam1*), including glomerular capillaries, peritubular capillaries, and larger vessels. APOL1 expression was not detected in proximal tubules or any other tubular segment.

None of the BAC-APOL1 mice spontaneously developed proteinuria, which is consistent with the original description of these mouse models [17, 18], but developed heavy proteinuria when intercrossed with a model of HIV-associated nephropathy (**Fig 5**). Using immunofluorescence, non-specific anti-APOL1 antibody immunostaining was observed in the wildtype and HIVAN mouse kidneys, mostly in Bowman capsule (**Fig 6A**). In the intercrossed mice with proteinuria, APOL1 protein was evident in glomeruli, but no APOL1 protein was detected in filtrate or within proximal tubules (**Fig 6B**). By Western blotting, APOL1 could not be detected in voided urine of proteinuric mice (data not shown). There also was no difference in the pattern of *APOL1* expression between the proteinuric APOL1-G0, -G1, or -G2 expressing mice. These same mouse kidneys were immunostained for APOA1, an apolipoprotein that is filtered, as a positive control for vascular distribution and proximal tubule reabsorption

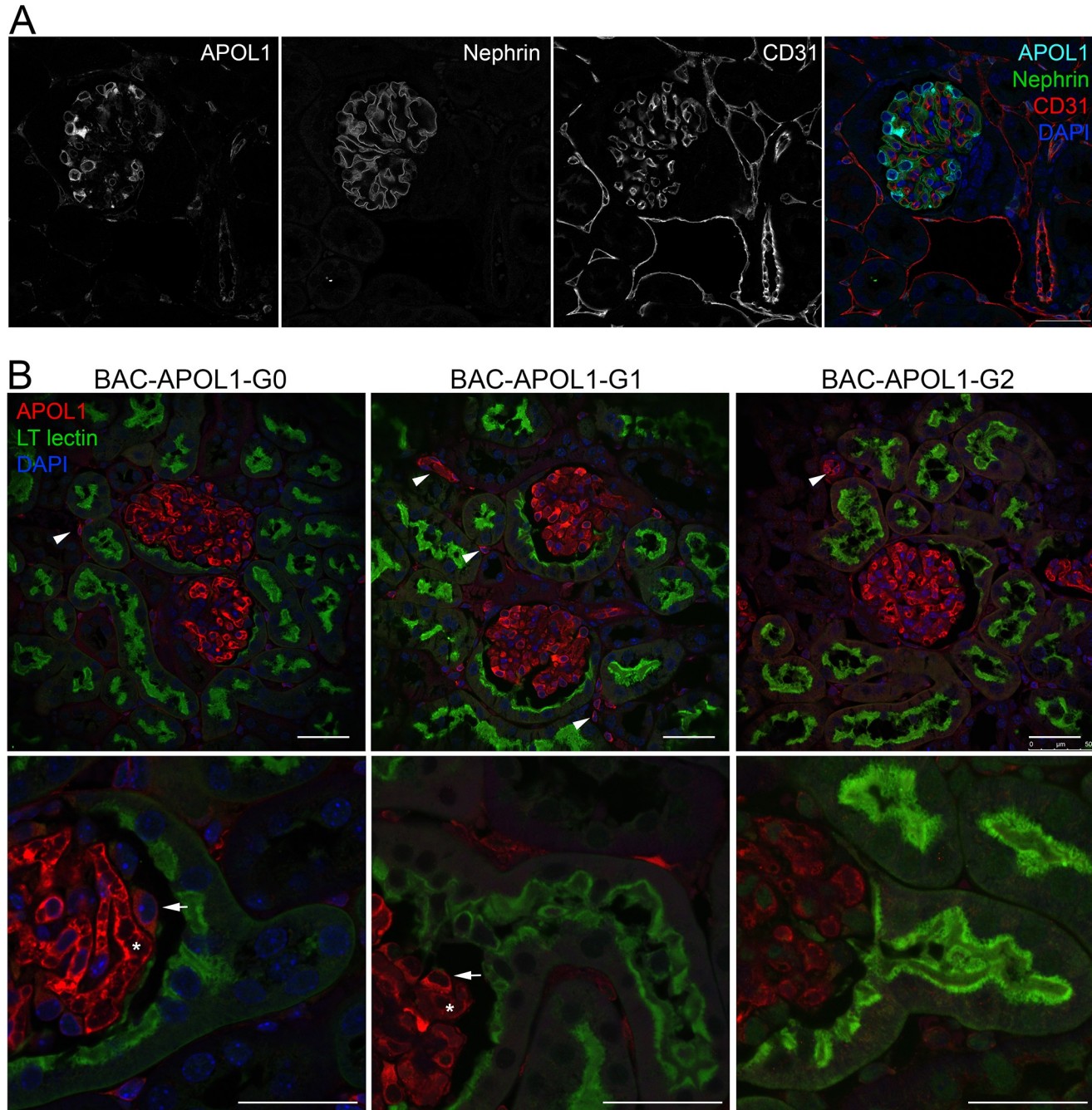

**Fig 2. APOL1 protein expression in BAC-APOL1 transgenic mouse kidneys. A**. Immunofluorescent staining for APOL1, nephrin (to identify podocytes), and CD31 (to identify endothelia cells) in the BAC-APOL1 transgenic mouse kidney (G1 mouse is shown). **B.** Comparison of APOL1 expression patterns in all three APOL1 transgenic lines. Proximal tubules were identified by labeling with fluorescent *Lotus tetragonolobus* (LT) lectin. APOL1 was present in vascular endothelia (arrow heads), in podocytes (arrows), and trapped in vascular spaces (*), but not tubular epithelia. Scale bar = 40µm.

patterns. Similar to APOL1, APOA1 in plasma was readily detected in glomerular capillary lumens. However, since APOA1 is filtered it also was present in protein reabsorption droplets at the proximal tubule brush border (**Fig 6C**). In the setting of proteinuria, APOA1-containing

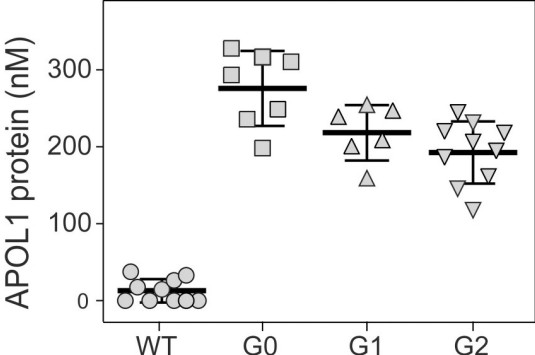

**Fig 3. APOL1 protein in circulation in the BAC-APOL1 transgenic mouse models.** Plasma APOL1 protein levels were measured by immunoassay in transgenic mice compared to age-matched wild-type (WT) littermates (WT, n = 10; G0, n = 6; G1, n = 7; G2, n = 10; all animals were male, approximately 21 weeks of age). APOL1 expression levels in each transgenic line were all significantly different than WT ($P < 0.0001$). Data are mean±SD, one measurement per animal, with significance determined by one-way ANOVA.

reabsorption droplets increased in number and size at the proximal tubule brush border. A similar pattern in the proximal tubule was not observed with APOL1.

## Discussion

Antibody specificity has been recognized as one of the most significant challenges impacting reproducible research [23]. We and others had originally reported APOL1 protein is present in proximal tubules of normal and diseased subjects [8, 9]. However, our continuing work with other anti-APOL1 antibodies and using mRNA *in situ* hybridization indicated *APOL1* was not expressed in proximal tubules [10]. A remaining possibility is that the APOL1 protein in blood is filtered and reabsorbed, resulting in the appearance of APOL1 protein in proximal tubules. Using *in situ* methods that do not rely on antibodies, along with the newly available human BAC-APOL1 transgenic mice, neither APOL1 protein nor *APOL1* mRNA could be detected in tubular epithelia. In the setting of proteinuria, APOL1 also was not filtered. In our hands, the Sigma rabbit polyclonal antibody consistently recognizes human APOL1 but has lot-to-lot differences with recognition of other proteins. The Sigma rabbit polyclonal lot used in studies here does not replicate the strong proximal tubule staining observed in prior lots sold under the same catalog number. Although monoclonal antibodies would eliminate the variation inherent in polyclonal antibody production lots, the commercial monoclonal antibodies tested had significant non-specificity or poor sensitivity for APOL1. A recent report describing the development and validation of a large number of monoclonal antibodies against human APOL1 also did not identify proximal tubule staining in human kidney [24].

Evidence against proximal tubule expression of *APOL1*, or a role for the proximal tubule in the APOL1-associated CKDs, is accumulating from several other groups. A study using similar humanized *APOL1* transgenic mouse models created with fosmids also did not find *APOL1* expression in the proximal tubule using both protein and mRNA detection methods [25]. Use of transgenic models with inducible *APOL1* expression that restrict expression to either podocytes or tubules found proteinuria and renal pathology occurs only when *APOL1* is expressed in podocytes, and not in the tubule [6]. As corroborating evidence in humans, several studies surveying the human urine proteome [26–30] did not identify APOL1 protein in urine, also indicating APOL1 is not filtered in any detectable quantity.

Our studies do not rule-out the possible release of small amounts of APOL1 into the primary filtrate from podocytes either by secretion or passively release when a podocyte dies and

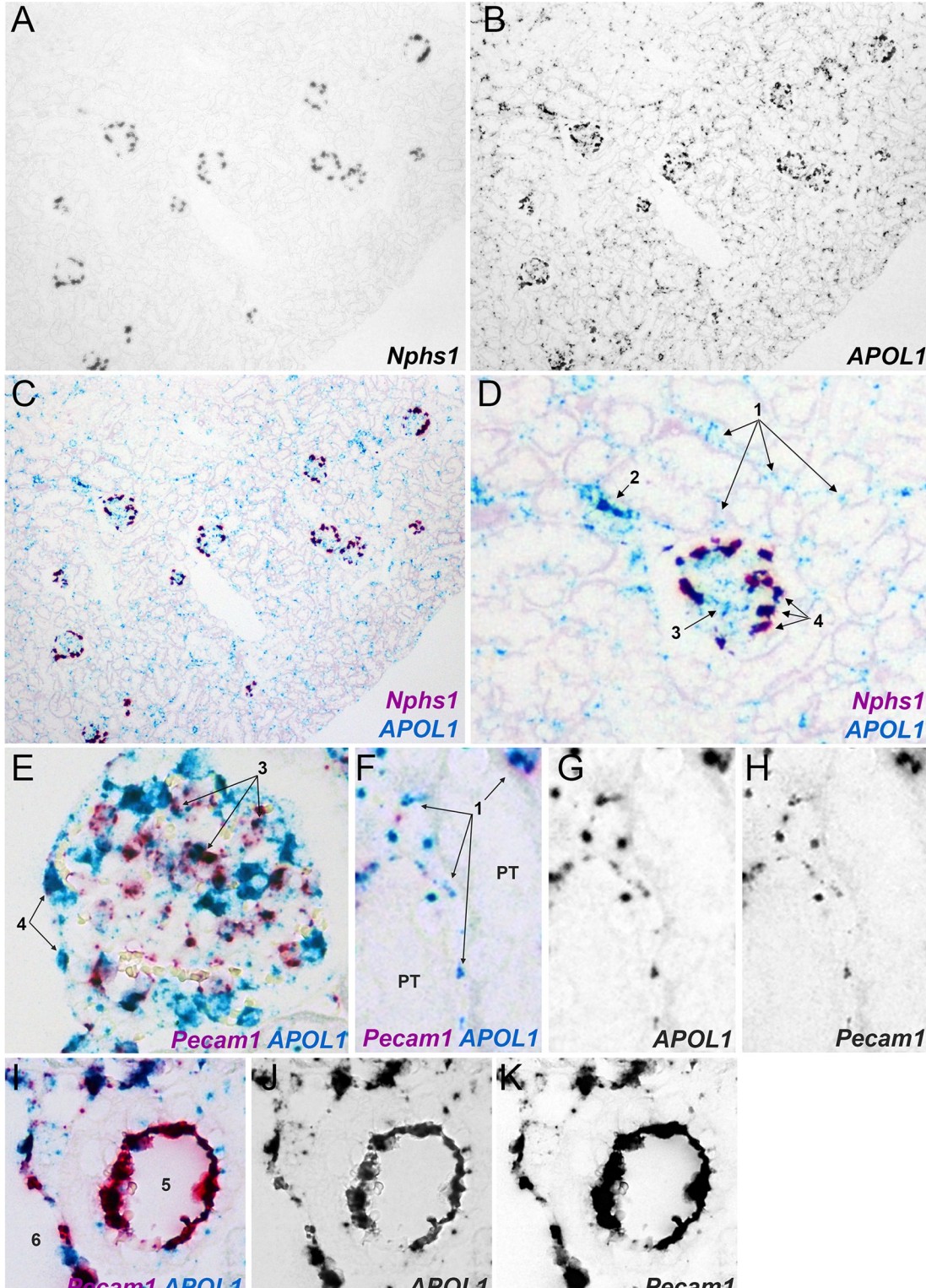

**Fig 4. *APOL1* mRNA expression in the BAC-APOL1 transgenic mouse models.** Duplex mRNA *in situ* hybridization for (A-D) *APOL1* and nephrin (*Nphs1*) gene expression and (E-K) *APOL1* and CD31 (*Pecam1*) expression to identified podocytes or endothelial cells respectively. *APOL1* expression was detected in peritubular capillaries ("1" arrows), arterioles ("2" arrows), glomerular capillaries ("3" arrows), podocytes ("4" arrows), and larger (interlobular) arteries ("5") and veins ("6") but not proximal tubules ("PT") or any other tubular segment. Images shown are from BAC-APOL1-G0 and BAC-APOL1-G1 mice.

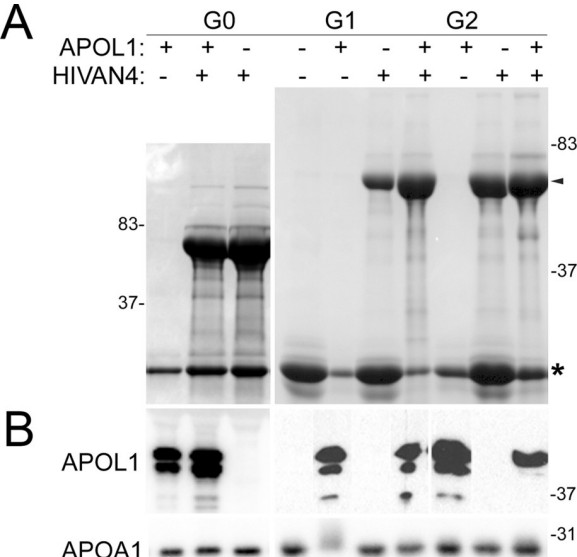

**Fig 5. BAC-APOL1 transgenic mouse models developed similar proteinuria when intercrossed with the HIVAN mouse model. A**. Example of proteinuria in single and dual transgenic mice assessed by gel electrophoresis of urine (Coomassie stain). Arrowhead marks albumin, and additional low molecular weight urinary proteins (asterisk) are a normal finding in mice. **B**. Western blot of mouse serum in the same single and dual transgenic mice showing maintenance of high levels of serum APOL1 protein in the setting of proteinuria. APOA1 Western blot as a control for a common serum protein that is freely filtered. Representative blot is shown; number of animal examined in each genotype group were: wildtype, n = 3; *HIVAN4*, n = 6; G0 x *HIVAN4*, n = 6; G1 x *HIVAN4*, n = 3; G2 x *HIVAN4*, n = 4.

lyses. Unlike the APOL1 expressed and secreted by hepatocytes, the major *APOL1* transcript in podocytes does not have a complete signal peptide [15], and there is no conclusive evidence that podocytes secrete APOL1. Alternatively, some studies observed the disease-induced high levels of *APOL1* can also produce rare alternatively spliced isoforms with different signal peptide sequences possibly permitting secretion [24, 31–33]. These potential alternative sources of APOL1 in filtrate would have been below limits of detection in the assays we and other investigators have used in kidney tissue and urine, and would be an unlikely explanation for the robust *APOL1* expression previously reported in proximal tubules. In addition, it is unclear if this potential low level of podocyte-derived APOL1 in filtrate would be of physiologic significance, considering that in humans, basal levels of *APOL1* expression are not associated with CKD. Our studies also cannot rule-out possible differences in *APOL1* expression between humans and the BAC-APOL1 transgenic mice. The function of podocytes and proximal tubules in blood filtration and reabsorption are fundamentally similar between humans and mice, however there are acknowledged limitations of using mice with regards to replicating human glomerular kidney diseases [34].

The work presented here and other published studies have shown significant similarities between humans [8–10, 24] and transgenic mouse *APOL1* expression patterns [17, 18, 24]. A reproducible and consistent observation from these combined studies is expression of *APOL1* in podocytes and in endothelial cells of glomerular capillaries, peritubular capillaries, and larger blood vessels. In addition, observations here indirectly support conclusions from prior studies [4, 5] that circulating APOL1 protein is unlikely to contribute to kidney disease pathogenesis as it is not the source of kidney-localized APOL1. Evaluating the contribution of podocyte- and endothelial-expressed *APOL1* is a logical focus for future studies examining the mechanism for *APOL1* risk variant contributions to CKD pathogenesis.

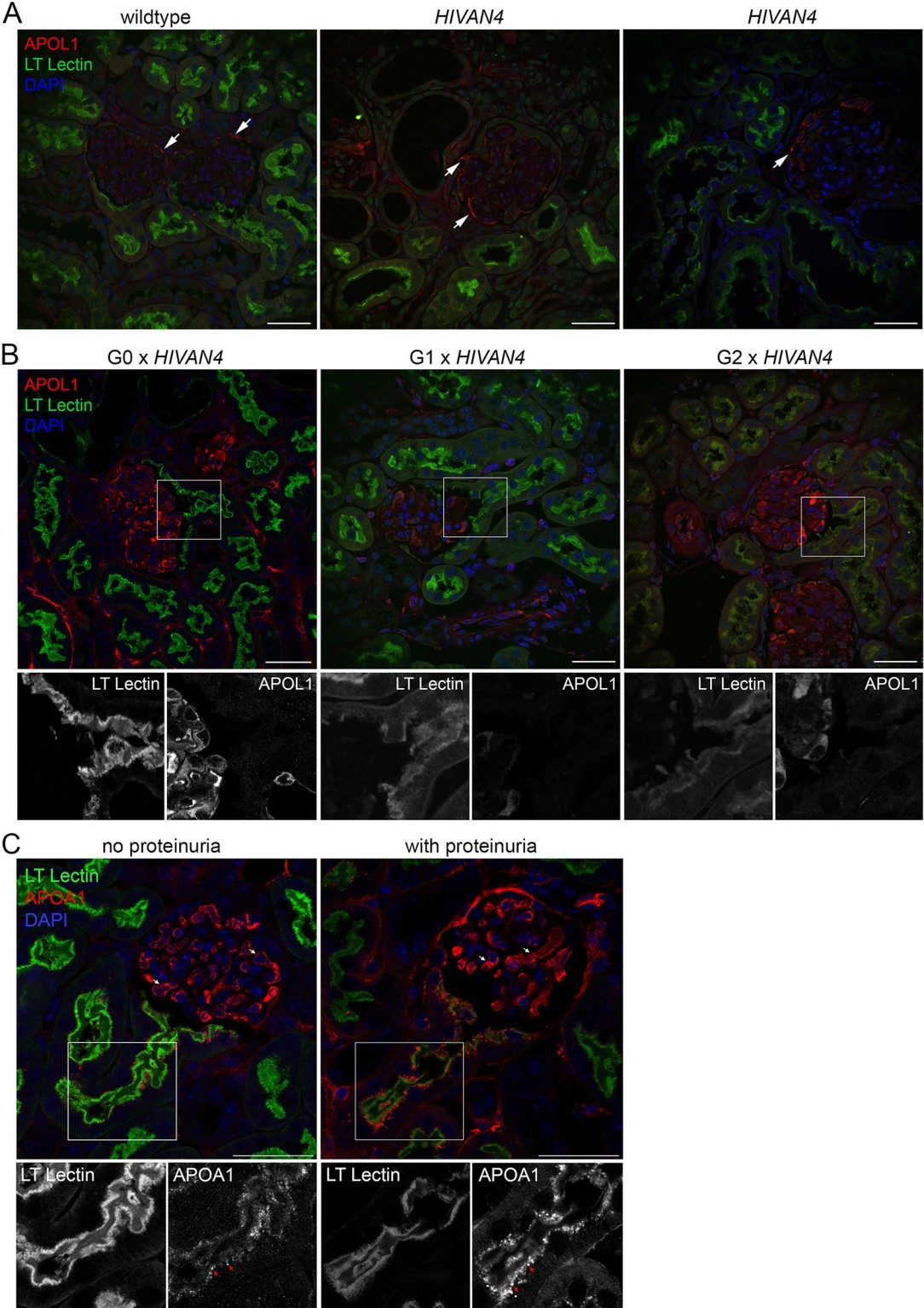

**Fig 6. APOL1 does not appear in proximal tubules in proteinuric BAC-APOL1 transgenic mice. A**. Control immunostaining in non-transgenic (wildtype) and proteinuric *HIVAN4* mice for APOL1 along with fluorescent *Lotus tetragonolobus* ("LT") lectin binding to demarcate the proximal tubule brush border. Since wildtype and *HIVAN4* mice do not have APOL1, the staining observed in parietal cells (arrows) is artifact. **B**. Immunostaining for APOL1 in proteinuric BAC transgenic mice of each *APOL1* genotype (representative images are shown, number of animal examined in each genotype

group were the same as for **Fig 5**). Images show proximal tubules at the transition with Bowman capsule. The boxed region is magnified below each panel along with the isolated fluorescent channels shown in black and white. **C**. Positive control immunostaining for a filtered lipoprotein, APOA1, and fluorescently-labelled *Lotus tetragonolobus* ("LT") lectin. An APOL1-G0 mouse and an APOL1-G0 x *HIVAN4* dual transgenic mouse with proteinuria is shown; below each respective color panel is the individual fluorescent channels (in black and white) of the boxed region for either LT lectin or APOA1. White arrows mark glomerular capillaries containing circulating APOA1 protein within capillary lumens, red arrows denote APOA1 in protein reabsorption droplets at the brush border of proximal tubules. Scale bar = 40μm.

## Supporting information

**S1 Data. Supplemental figures and tables.**
(PDF)

**S1 Raw images. Original, uncropped gel images.**
(PDF)

## Acknowledgments

We thank William Baldwin and Nina Dvorina for assistance with the antibody validation studies.

## Author Contributions

**Conceptualization:** Sethu M. Madhavan, Myung K. Shin, Maarten Hoek, John R. Sedor, John F. O'Toole, Leslie A. Bruggeman.

**Data curation:** Jonathan W. Choy, Michelle Chen, Myung K. Shin, Leslie A. Bruggeman.

**Formal analysis:** Sethu M. Madhavan, Jonathan W. Choy, Michelle Chen, Myung K. Shin, Maarten Hoek, John R. Sedor, John F. O'Toole, Leslie A. Bruggeman.

**Funding acquisition:** John R. Sedor, John F. O'Toole, Leslie A. Bruggeman.

**Investigation:** Natalya A. Blessing, Zhenzhen Wu, Sethu M. Madhavan, Jonathan W. Choy, Michelle Chen, Leslie A. Bruggeman.

**Methodology:** Zhenzhen Wu, Sethu M. Madhavan, Jonathan W. Choy, Michelle Chen, Leslie A. Bruggeman.

**Project administration:** Myung K. Shin, Maarten Hoek, John R. Sedor, John F. O'Toole, Leslie A. Bruggeman.

**Resources:** Myung K. Shin.

**Supervision:** Myung K. Shin, Maarten Hoek, John R. Sedor, John F. O'Toole, Leslie A. Bruggeman.

**Validation:** Myung K. Shin.

**Writing – original draft:** Natalya A. Blessing, Leslie A. Bruggeman.

**Writing – review & editing:** Zhenzhen Wu, Sethu M. Madhavan, Jonathan W. Choy, Michelle Chen, Myung K. Shin, Maarten Hoek, John R. Sedor, John F. O'Toole, Leslie A. Bruggeman.

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
