## [Decision Letter · Decision Letter 0]

19 Mar 2021

PONE-D-21-04749

APOL1 is not expressed in proximal tubules and is not filtered

PLOS ONE

Dear Dr. Bruggeman,

Thank you for submitting your manuscript to PLOS ONE. After careful consideration, we feel that it has merit but does not fully meet PLOS ONE’s publication criteria as it currently stands.  At the same time I note that all three reviewers were enthusiastic about the project and the conclusions.

Reviewer 2 made several specific technical critiques about the work.  I would like the authors to pay special attention to the critiques of the various co-localization studies (a point that was echoed by Reviewer 3). Those analyses are central to the argument made in this MS. I agree that it might be helpful to show more micrographs, perhaps at different magnifications and with more glomeruli in the field. The authors should also add some text to address the issue that ApoL1 expression in this somewhat artificial transgenic mouse model looks a bit different than it does in humans. It would also be helpful to address the comment about the Santa Cruz antibody mentioned by the reviewer compared to the one that was used in this study, at least in the response to critiques if not in the revised MS itself. It is also important to be transparent about all statistical methods used.  Reviewer 3 also raises some other technical issues that I think will be quite straightforward to address, and that may simply be the way the figures were prepared.  That reviewer also makes some useful suggestions on spelling and formatting. 

We look forward to receiving your revised manuscript.

Kind regards,

Stuart E Dryer, PhD

Academic Editor

PLOS ONE

Journal Requirements:

2. Please provide your probe sequences (or catalog numbers) for your  in situ hybridization probes."

3. Please include further information regarding your in vivo study, per our guidelines (http://journals.plos.org/plosone/s/submission-guidelines#loc-animal-research). Specifically, please provide details regarding:

 a) Animal health monitoring, including:

     -frequency of monitoring,

     -monitoring criteria, and

     -any efforts made to reduce suffering and distress, such as administering

      analgesics.

 b) whether humane endpoints were in place during the study and how they were applied,

 c) the method of euthanasia for the animals,

 d) any mortality that occurred outside of planned euthanasia or humane endpoints,

 e) the source of all of the animals, and

 f) The total number of animals."

4. Thank you for stating in the text of your manuscript "All animal studies were reviewed and approved by Institutional

Animal Care and Use Committees at the Cleveland Clinic and Case Western Reserve University." Please also add this information to your ethics statement in the online submission form."

6.  Thank you for stating the following in the Competing Interests section:

We have read the journal's policy and the authors of this manuscript have the following competing interests: The authors MH, JC, MC, and MKS were employees of Merck & Company, Inc. during the conduct of this study. 

We note that one or more of the authors are employed by a commercial company: Merck & Company & Maze Therapeutics.

Additional Editor Comments (if provided):

Reviewers' comments:

Reviewer's Responses to Questions

**Comments to the Author**

1. Is the manuscript technically sound, and do the data support the conclusions?

Reviewer #1: Yes

Reviewer #2: Partly

Reviewer #3: Yes

2. Has the statistical analysis been performed appropriately and rigorously? 

Reviewer #1: Yes

Reviewer #2: No

Reviewer #3: N/A

3. Have the authors made all data underlying the findings in their manuscript fully available?

Reviewer #1: Yes

Reviewer #2: Yes

Reviewer #3: Yes

4. Is the manuscript presented in an intelligible fashion and written in standard English?

Reviewer #1: Yes

Reviewer #2: Yes

Reviewer #3: Yes

5. Review Comments to the Author

Reviewer #1: Bruggeman et al studied APOL1 expression in tubular cells and its filtration in the tubular lumen.

Comments:

1. These investigators were pioneer to show expression of APOL1 protein in kidney cells including tubular cells. However, they have now clarified that tubular cells do not express APOL1.

2. Interestingly, they have also shown that it is not filtered. However, a possibility of its secretion by podocytes can not be excluded.

Reviewer #2: This current study tested if ApoL1 is synthesized and/or reabsorbed by proximal tubule cells. Expression of ApoL1 mRNA and protein by renal tissues of human and ApoL1 transgenic mice were examined using in situ hybridization, immunofluorescence and immunohistochemistry techniques with commercially available antibodies. The authors concluded that ApoL1 is mainly expressed in podocytes and endothelial cells, but not by tubular cells.

Overall, the findings in this study are informative, but not novel. The manuscript is logically well written, and the data is straightforward. However, supporting evidence seems insufficient because all data except for Fig 1 was generated from a transgenic mouse model. Also, the data cited that corresponds to the conclusions made regarding the filtration and reabsorption of ApoL1 part seems weak.

Comments:

1. Most studies relied on the results from BAC-ApoL1 Tg animals, in which the expression and localization of ApoL1 most likely is different from that in humans.

Indeed, mRNA and protein expression patterns of ApoL1 in liver of human and Tg mouse (Supp F2) seem quite different (disperse vs focal). Thus, human renal tissues from healthy and diseased individuals need to be thoroughly tested using the Santa Cruz antibody (lot E105260).

2. Immunofluorescence on human kidneys (Fig 1A) shows a single glomerulus with limited views of tubules. Please show the ApoL1 protein expression pattern from more tubular regions following co-staining with proximal tubule cell markers.

3. In Fig 4 (ISH on BAC-ApoL1 Tg mouse, G0? G1? G2?), decent amounts of ApoL1 mRNA expression were also observed in Nphs1 negative cells. To make it clear that the ApoL1 signals are definitively not from PT, duplex mRNA ISH for ApoL1 and PT markers need to be performed.

4. The authors showed that ApoA1 but not ApoL1 is colocalized with LT-lectin, a tubule marker (Fig 6). However, it is still not clear if positive staining of ApoA1 observed in tubular cells resulted from reabsorption from the filtrate or if it is expressed by tubular cells. Adequate measurements need to be done to clarify this issue.

We cannot rule out the possibility that a small amount of ApoL1 is released from damaged/apoptotic podocytes and reabsorbed by the proximal tubules from the filtrate.

5. If possible, additional examination of ApoL1 protein expression in human and mouse urine samples (G0, G1, and G2 genotype) would be informative.

6. Also, in Supp F1, it would be informative if the authors show the ApoL1 staining pattern in ApoL1 Tg and normal mouse renal tissues using the Santa Cruz antibody (lot E105260).

7. The statistical method used in Fig 3 has not been described.

8. The conclusions of the paper, as stated in the title is overly quite bold. The authors fall short in providing definitive evidence for the statement that proximal tubule cells neither produce nor reabsorb ApoL1 and should therefore tone down their conclusions.

Reviewer #3: Bruggeman and colleagues report that APOL1 is not expressed in proximal tubules and is not filtered. The investigators have studied (Merck) BAC/APOL1 mice, representing each of the three genotypes, crossed to FVB/N mice for 10 generations. Their prior in situ hybridization work suggests that RNA expression is confined to podocytes and endothelial cells in glomerular and peritubular capillaries, and other vessels (reference 10).

Here, the authors have done a careful analysis of various APOL1 antibodies, particularly of the Sigma polyclonal antibody. The quality of the immunostaining and in situ hybridization images is excellent. The authors conclude that bona fide APOL1 protein expression in the BAC transgenic mice is limited to podocytes and endothelial cells. They suggest that false-positivity for APOL1 immunostaining in the proximal tubule may be due to glycosylation patterns. These finding will be of value to investigators in this field.

The paper is clear, concise and elegantly written.

1) Figure 1. To my eye, it appears that there is only partial co-localization of GLEPP1 (perhaps located in peripheral portions of glomerular capillary loops) and APOL1 (possibly in the cell bodies). I wonder whether the authors agree. I agree that both proteins are in podocytes as well as other cells. I think all this might be easier to resolve if the images were larger.

2) Figure 5, Western, is confusing, because two blots were but together, with result that order of the lanes differ and protein migration differs. A single blot would strengthen the paper.

3) After all the effort to cross the mice for 10 generations onto an FVB/N background, it would be useful to know whether the kidney phenotype become more severe (more proteinuria, more glomerulosclerosis).

Minor comments.

P4,P8, capitalize Coomassie, as it is (or rather was) a place name.

P5, capitalize Lotus, as it is a genus name

PP6-7. “APOL1 expression was not detected in proximal tubules or any other nephron segment.” This sentence appears twice. More importantly, the glomerulus is part of the nephron. Suggest “or any other tubular segment.”

P7, no need to capitalize nephrin

P8, Lotus tetragonolobus, capitalize genus, italics

Bowman’s capsule > Bowman capsule. The trend is away from the possessive, as the discoverer does not own the structure. It does sound odd to say Bowman capsule after we have been saying it differently for decades.

6. PLOS authors have the option to publish the peer review history of their article (what does this mean?). If published, this will include your full peer review and any attached files.

Reviewer #1: **Yes: **Pravin C. Singhal

Reviewer #2: No

Reviewer #3: No

---

## [Author Response · Author response to Decision Letter 0]

11 May 2021

Editorial comments:

1. PLOS ONE style requirements are met.

2. Probe catalog numbers for in situ hybridization probes have been included.

3. Detailed information on the in vivo study is included. 

4. Ethics statement has been revised and is in the cover letter and the methods section. 

5. Original uncropped images are now included in the supplemental file.

6. A revised funding and competing statement is in the cover letter, and the author roles are correct in the author contributions section. 

7. Reference list is complete and correct.

Reviewer critiques:

Reviewer #1

Interestingly, they have also shown that it is not filtered. However, a possibility of its secretion by podocytes cannot be excluded.

 RESPONSE: We agree this remains a possibility and have included a paragraph in the discussion to address this issue.

Reviewer #2

1. Most studies relied on the results from BAC-ApoL1 Tg animals, in which the expression and localization of ApoL1 most likely is different from that in humans. Indeed, mRNA and protein expression patterns of ApoL1 in liver of human and Tg mouse (Supp F2) seem quite different (disperse vs focal). Thus, human renal tissues from healthy and diseased individuals need to be thoroughly tested using the Santa Cruz antibody (lot E105260).

 RESPONSE: We believe the reviewer is referring to the Sigma – not Santa Cruz – antibody as Santa Cruz antibodies were not used and the lot number given is for the Sigma antibody. A major point of this manuscript is to highlight the limitation of antibody reagents, and one significant issue regards polyclonal antibody lots, which when exhausted are gone and studies can never be repeated or reproduced. Unfortunately, lot E105260 is now exhausted both in our lab and is no longer sold by Sigma (we have revised supplemental Table 1 to indicate this). We cannot do additional studies that require this antibody lot. 

 The comparison of mouse and human liver we agree there are some differences. It should be noted that the mouse livers were from normal young adults, but the human livers were “normal” margins of cancer resections and also had pathologic changes consistent with steatosis. We have revised the description of the liver expression pattern in the text and figure legends to be more specific on these issues. In general, we concur that mice and humans are different on many levels, and there may be issues studying a human gene in mice. We agree we should acknowledge this as a limitation of our study, and have included a discussion of this issue relevant to our observations.

2. Immunofluorescence on human kidneys (Fig 1A) shows a single glomerulus with limited views of tubules. Please show the ApoL1 protein expression pattern from more tubular regions following co-staining with proximal tubule cell markers.

 RESPONSE: As stated above, we cannot do additional immunostaining with tubule makers, as this APOL1 antibody is no longer available. 

3. In Fig 4 (ISH on BAC-ApoL1 Tg mouse, G0? G1? G2?), decent amounts of ApoL1 mRNA expression were also observed in Nphs1 negative cells. To make it clear that the ApoL1 signals are definitively not from PT, duplex mRNA ISH for ApoL1 and PT markers need to be performed.

 RESPONSE: We have examined all three genotypes (G0, G1, and G2) of the BAC-APOL1 mice using ISH and have added this statement to the results section, and also indicated which genotype is shown in the figure. We have added to Figure 4 new data on duplex mRNA ISH for CD31 (Pecam1). The extra-glomerular staining is peritubular capillaries, not proximal tubules, so we used an endothelial marker as a cell-type specific marker to co-label with extraglomerular APOL1-expressing cells. 

4. The authors showed that ApoA1 but not ApoL1 is co-localized with LT-lectin, a tubule marker (Fig 6). However, it is still not clear if positive staining of ApoA1 observed in tubular cells resulted from reabsorption from the filtrate or if it is expressed by tubular cells. Adequate measurements need to be done to clarify this issue. We cannot rule out the possibility that a small amount of ApoL1 is released from damaged/apoptotic podocytes and reabsorbed by the proximal tubules from the filtrate.

 RESPONSE: APOA1 is expressed only by the liver, as such, we will not be able to show endogenous APOA1 expression in tubule cells, since no resident kidney cell expresses APOA1. The second point of possibly small amounts of APOL1 being released into filtrate from a dead podocyte and reabsorbed by the tubule remains plausible, however, this small amount would not account for the robust staining previously reported for the proximal tubule. We have added a paragraph to the discussion section describing other possible sources of APOL1 in the primary filtrate.

5. If possible, additional examination of ApoL1 protein expression in human and mouse urine samples (G0, G1, and G2 genotype) would be informative.

 RESPONSE: We had examined urine from the HIVAN intercrossed BAC mice by Western blotting and found no evidence of APOL1 in urine. We have included this in the results section. 

6. Also, in Supp F1, it would be informative if the authors show the ApoL1 staining pattern in ApoL1 Tg and normal mouse renal tissues using the Santa Cruz antibody (lot E105260).

 RESPONSE: Again, we think the reviewer is referring to the Sigma (not Santa Cruz) antibody here. As per point #1, we cannot do additional studies that require lot E105260, as the supply is exhausted and is no longer sold by Sigma.

7. The statistical method used in Fig 3 has not been described.

 RESPONSE: This has been added to the figure legend (ANOVA).

8. The conclusions of the paper, as stated in the title is overly quite bold. The authors fall short in providing definitive evidence for the statement that proximal tubule cells neither produce nor reabsorb ApoL1 and should therefore tone down their conclusions.

 RESPONSE: We have revised the title and conclusions, taking into consideration the above comments. 

Reviewer #3

1. Figure 1. To my eye, it appears that there is only partial co-localization of GLEPP1 (perhaps located in peripheral portions of glomerular capillary loops) and APOL1 (possibly in the cell bodies). I wonder whether the authors agree. I agree that both proteins are in podocytes as well as other cells. I think all this might be easier to resolve if the images were larger.

 RESPONSE: We have revised Figure 1 to include higher magnification images. In our prior studies of human APOL1 expression patterns (Refs 8, 10), we demonstrated that podocyte markers such as GLEPP1 reside in a different subcellular location than APOL1. Consequently, there is not a 100% overlap in fluorescent signals. We are using proteins such as GLEPP1 for cell identity, not protein co-localization with APOL1. In addition to podocytes, some of this signal is attributable to glomerular endothelial cells which are also positive for APOL1 (CD31 as a cell identifying marker). We have also added the mRNA in situ images with CD31 (PECAM1) co-staining for endothelial cells in Figure 4. 

2. Figure 5, Western, is confusing, because two blots were but together, with result that order of the lanes differ and protein migration differs. A single blot would strengthen the paper.

 RESPONSE: We agree, but we are working with spot mouse urines and had limited volumes from this cohort of intercrossed mice. We have no additional material for re-running gels.

3. After all the effort to cross the mice for 10 generations onto an FVB/N background, it would be useful to know whether the kidney phenotype become more severe (more proteinuria, more glomerulosclerosis)

 RESPONSE: We understand this comment, as it is well-known that many glomerular injury models are significantly different in phenotype based on the mouse genetic background. All BAC-APOL1 (G0, G1, G2) mice have no renal phenotype either on the original background (129SvJ) or the FVB/N background. There is no difference in the APOL1 phenotype on the two mouse strains in the absence of a disease-inducing stressor. To induce significant proteinuria, we intercrossed the BAC mice with the HIVAN model, and the backcrossing to the FVB/N background was done specifically to conduct the HIVAN crossbreeding. The phenotype of the HIVAN mouse model is highly dependent on the genetic strain, which we have already examined in detail (PMIDs 21784893, 19381020, 14983036). We are currently conducting a comprehensive, longitudinal study of disease phenotype comparisons based on APOL1 genotype in similar HIVAN intercrosses, but study outcomes are many months from completion, and beyond the scope of this manuscript. 

Minor comments. 

 RESPONSE: All spelling and wording issues have been corrected as recommended.

---

## [Decision Letter · Decision Letter 1]

31 May 2021

Lack of APOL1 in proximal tubules of normal human kidneys and proteinuric APOL1 transgenic mouse kidneys.

PONE-D-21-04749R1

Dear Dr. Bruggeman,

We’re pleased to inform you that your manuscript has been judged scientifically suitable for publication and will be formally accepted for publication once it meets all outstanding technical requirements.

Kind regards,

Stuart E Dryer, PhD

Academic Editor

PLOS ONE

Additional Editor Comments (optional):

Reviewers' comments:

Reviewer's Responses to Questions

**Comments to the Author**

1. If the authors have adequately addressed your comments raised in a previous round of review and you feel that this manuscript is now acceptable for publication, you may indicate that here to bypass the “Comments to the Author” section, enter your conflict of interest statement in the “Confidential to Editor” section, and submit your "Accept" recommendation.

Reviewer #2: All comments have been addressed

Reviewer #3: All comments have been addressed

2. Is the manuscript technically sound, and do the data support the conclusions?

Reviewer #2: Yes

Reviewer #3: Yes

3. Has the statistical analysis been performed appropriately and rigorously? 

Reviewer #2: Yes

Reviewer #3: Yes

4. Have the authors made all data underlying the findings in their manuscript fully available?

Reviewer #2: Yes

Reviewer #3: (No Response)

5. Is the manuscript presented in an intelligible fashion and written in standard English?

Reviewer #2: Yes

Reviewer #3: Yes

6. Review Comments to the Author

Reviewer #2: In this revised version, authors addressed my concerns and made the corrections accordingly. I have no substantive suggestions to further improve the manuscript.

Reviewer #3: The authors have addressed all the issues raised. This is an important paper. Congratulations on the fine work.

7. PLOS authors have the option to publish the peer review history of their article (what does this mean?). If published, this will include your full peer review and any attached files.

Reviewer #2: No

Reviewer #3: **Yes: **Jeffrey Kopp

---

## [Editor Report · Acceptance letter]

9 Jun 2021

PONE-D-21-04749R1 

Lack of APOL1 in proximal tubules of normal human kidneys and proteinuric *APOL1* transgenic mouse kidneys. 

Dear Dr. Bruggeman:

I'm pleased to inform you that your manuscript has been deemed suitable for publication in PLOS ONE. Congratulations! Your manuscript is now with our production department. 

Kind regards, 

on behalf of

Dr. Stuart E Dryer 

Academic Editor

PLOS ONE